# Reproductive health needs of Human papillomavirus (HPV) positive women: A systematic review

**Mina Galeshi[1], Hoda Shirafkan[2], Shahla Yazdani[3], Zahra Motaghi[4]\***

1 Student Research Committee, School of Nursing and Midwifery, Shahroud University of Medical Sciences, Shahroud, Iran, 2 Social Determinants of Health Research Center, Health Research Institute, Babol University of Medical Sciences, Babol, Iran, 3 Cancer Research Center, Health Research Institute, Babol University of Medical Sciences, Babol, Iran, 4 Department of Reproductive Health, School of Nursing and Midwifery, Shahroud University of Medical Sciences, Shahroud, Iran

* galeshi_m@yahoo.com

**Data Availability Statement:** All relevant data are within the paper and its Supporting Information file. However, any question or other file data is required you can contact us using the email address galeshi_m@yahoo.com"

## Abstract

### Objectives

Human papillomavirus is one of the most important causes of cervical cancer. Participating in an HPV test and receiving an HPV diagnosis can create questions about the needs and preferences. The present systematic review was conducted to determine the reproductive health needs of women with HPV.

### Methods

We searched PubMed, Scopus, Web of Science, Google Scholar and Magiran, SID and Iranmedex. Without language restrictions and time constraints. We also searched the grey literature and carried out forward/backward citation searches.

### Results

In the first, 1056 articles were retrieved, and, after removing them, 13 articles published were entered. The studies were qualitative (N = 9), quantitative (N = 3), and one was unclear. Most qualitative studies collected data using individual interviews (N = 7), two qualitative studies, narratives of HPV patients from a website of patient experiences and questions. Women wanted further information on different HPV viral types, transmission, implications for sexual partners, prevalence, latency and regression of HPV, their management options and the implications of infection for cancer risk and fertility. Women's experience of searching the Internet for further information about HPV was reported as difficult, anxiety provoking and contributing to the stigma of the infection because information was often located in the context of other sexually transmitted infections, with multiple sexual partners highlighted as a risk factor for infection.

### Conclusion

Surveys showed that the majority of women had unanswered questions about their HPV test results. The information that women thought was helpful in interpreting their test results

"shahla_yazdani_1348@yahoo.com", upon reasonable request.

**Funding:** The authors received no specific funding for this work.

**Competing interests:** The authors have declared that no competing interests exist.

**Abbreviations:** HPV, Human Papillomavirus; COCs, contraceptive pills; LNG, levonorgestrel.

included having a high-risk type of HPV, and cancer survival statistics for the virus. Women also needed information about sexual transmission, how HPV tested positive in a long-term relationship, and the potential consequences for their partners and the risk of re-infection. Younger women had questions about whether HPV could affect fertility.

## Introduction

Human papillomavirus (HPV) is one of the most important sexually transmitted viruses [1]. This virus is one of the causes of cervical cancer [2]. Although this cancer has many environmental and genetic causes (such as smoking, early marriage, high parity, etc.), the most important cause is infection with high risk types of HPV [3]. Cervical cancer is the second most common cancer in women, with 500,000 new cases reported each year [4]. Various and extensive studies have been conducted to investigate the prevalence of HPV infections in different countries as well as in Iran, which indicates its increasing prevalence in the world [5–7]. The prevalence of this virus in the world is between 10.5–55.4% and in Iran, it is 49.5% [6, 7]. The prevalence of genital HPV infection in the United States among women aged 18 to 59 years was 40% for all types of HPV and 20% for high-risk HPVs [8].

In general, cervical cancer is not only completely preventable but also completely curable if detected and treated in the early stages [9], and its prevalence is 6 times higher in developing countries; than in developed countries, cervical cancer is mainly due to differences and the lack of appropriate screening programs in these countries [10]. More than 95% of cervical cancers are attributed to papillomavirus [11]. More than 100 strains of the virus have been identified, but types 16 and 18 are the most important strains of the virus in developing cervical cancer [1]. Types 6 and 11 of this virus cause warts in the vulva [12], vagina, cervix, and penis [13].

This disease affects the quality of life of individuals and imposes financial costs on health systems [14]. It has been found that 35–65% of people become infected with HPV during sex [15]. Understanding cervical cancer, along with being involved with HPV infection, in patients with HPV raises questions about needs and preferences. In women undergoing cervical cancer screening, it is important to be aware of ways to minimize anxiety [16]. Previous research has been more limited to the field of HPV DNA testing [17]. HPV DNA testing included information on HPV disease, prevention, treatment, and risk of cancer [18].

Having an HPV test and getting the result can cause undesirable psychosocial reactions [19]. To prevent anxiety and psychological distress, education and counseling are increasingly important for women diagnosed with HPV [20]. Therefore, more attention should be paid to how to communicate and provide information [21].

Women with HPV face many communication challenges and information gaps in the stages of cervical cancer screening in primary care, waiting time for referral to a specialist, first consultation, and after consultation in specialized care [22]. Relatively little research has shown that health care informational challenges for women diagnosed with HPV and cervical intraepithelial neoplasia [23, 24].

Counseling with HPV patients poses special challenges for physicians and patients, especially in communities where HPV is rare and associated with severe stigma [25]. It will be invaluable to provide a deep understanding of the challenges, perspectives, experiences, and needs of women living with HPV positively at the primary and specialist care levels. Also HPV infection and subsequent cervical cancer are considered to be important public health problem

worldwide. Some reproductive factors are associated with increasing incidence of this infection. Therefore, improvement of our knowledge about reproductive factors associated with HPV may help us to identify women at risk and to developed different methods of preventive interventions. Therefore, the present systematic review was performed to determine the needs of women with HPV.

## Material and methods

The present study was approved by the Ethics Committee of Shahroud University of Medical Sciences: IR.SHE.REC.1400.154 and after receiving the code from the Prospero system Code: CRD42021293223 Based on the proposed systematic review and meta-analysis checklist (PRISMA), it was done.

### Search strategy for identifying papers

Related articles, with electronic search in medical databases such as: PubMed, SCOPUS, Web of Science, Google Scholar, search engine and Iranian Magiran, Scientific Information Database and Iranmedex and additional articles with a gray literature search using OpenGrey (www.opengrey.eu). We searched for articles without language, Geographical area and time restrictions.

### Selection process

As the reproductive health needs of women with HPV were not fully understood, a comprehensive and systematic study was conducted on the subject. Inclusion criteria include Studies related to the needs of women with HPV, conference-related abstracts, and exclusion criteria: Studies related to HPV in men, complications of HPV, HPV-related risk factors, cervical cancer, Treatment of cervical cancer or colposcopy, commentaries, opinion pieces, and editorials.

### Data extraction

All the articles obtained in the introductory search at first reviewed by title, then Abstract, and finally their full text. The articles were reviewed independently by two people. The names of the authors and the journal were not hidden from reviewers. Disagreements between reviewers were decided by negotiating with a third reviewer until a final agreement was reached. The data extracted for analysis including the author's name, year of the study, place of the study, study method, and the sample size in each group were entered into the electronic datasheet. Researchers contacted their authors for more information when they could not retrieve articles from authorized databases. All studies were reviewed for duplication. For each study, the data is listed in the table based on the study checklist.

### Quality assessment

A quantitative/qualitative adjusted checklist (CASP) was used to critically evaluate the articles. The Checklist (CASP) is a standard tool for evaluating articles, developed by the JAMA Group in 1994, and is the oldest and most widely used critical evaluation tool for a variety of articles/ studies [26].

The present checklist was 10 items to help understand qualitative research. How to use this tool for evaluating qualitative studies is as follows: When evaluating, three general issues should be considered first. A: Are the study results valid enough? B: What are the results? A: Will the results help locally? The first two questions are systematically screened and can be answered quickly. If the answer to both is "yes", the remaining questions are continued. Each item was

assigned a score of 2 (including that item in the article) or zero (not paying attention to the item in the article) and one (cannot be said) and the total score of this checklist was between 0–20.

To evaluate cross-sectional studies, this tool consisted of 18 items, and each item was given a score of one. Including that item in the article (or zero) was considered as not paying attention to the item in the article. Attitude evaluation (3 items), study design (5 items), and results (5 items) were divided and the total score of this checklist varied between 0–18.

After carefully reading the full text of each article, an article quality evaluation checklist was completed by the first researcher and items were scored. Reassessment was performed in the same way by the second researcher. If there was no agreement on scoring the items, the final score would be taken in a joint session.

In the next step, these articles were compared in terms of scores obtained in each of the four areas as well as in terms of total scores. Also, the qualitatively reviewed articles were divided into three categories of good, medium, and poor quality articles according to the scores obtained from this checklist. The total score is 20 high quality; 16–19 medium quality; ≤15 was considered low quality. Also, the articles reviewed cross-sectional in consultation with experts, scores of 75% of the total score and above as good quality (score≥ 13), scores between 25–25% of the total score as average quality (scores 6–12) and scores less than 25% of the total score (score 5 and below) were divided as poor quality [27].

### Analysis

Quantitative and qualitative findings were analyzed separately. The results of quantitative studies were reported descriptively by first creating an initial combination of findings, existing relationships and data strength were examined.

For qualitative studies, a thematic synthesis was performed by examining the text line by line in the results section and discussing and then identifying descriptive topics.

## Results and discussion

The initial search returned 1056 articles, which was reduced to 1020 after the removal of duplicates. Of these, 959 were excluded on the basis of their title, leaving 61 abstracts to be reviewed. Following exclusions, 48 full-texts were reviewed. 13 articles were excluded during the full-text review and an additional one were identified following backward and forward citation searches (Fig 1).

Studies were conducted in the United States (N = 5), Europe (N = 4), Asia (N = 4). The studies were mostly qualitative (N = 9), quantitative (N = 3), and one type of study was unclear. Most qualitative studies collected data using individual interviews (N = 7), two qualitative studies, narratives of HPV patients from a website of patient experiences and questions. The characteristics of the participants and the study are shown in the table (Table 1).

The results of the overall quality evaluation of these articles showed that 8 articles were of good quality, 2 were of average quality and one was of poor quality. There is not enough information about Friedman AL *et al*. and Garland SM articles to evaluate the quality, but since these studies were valuable, they were included in the study according to the opinion of the research team.

The results of the article evaluation are divided into four areas of the checklist; the total score of each article, and article quality were in (Tables 2 and 3).

### Summary of the HPV Woman's needs including

1. Signs, Symptoms, Transmitted, Cause, Consequences of HPV, 2. HPV and sexual partner, 3. Seeking needed information for women with HPV, 4. HPV and social consequences, 5. Needs

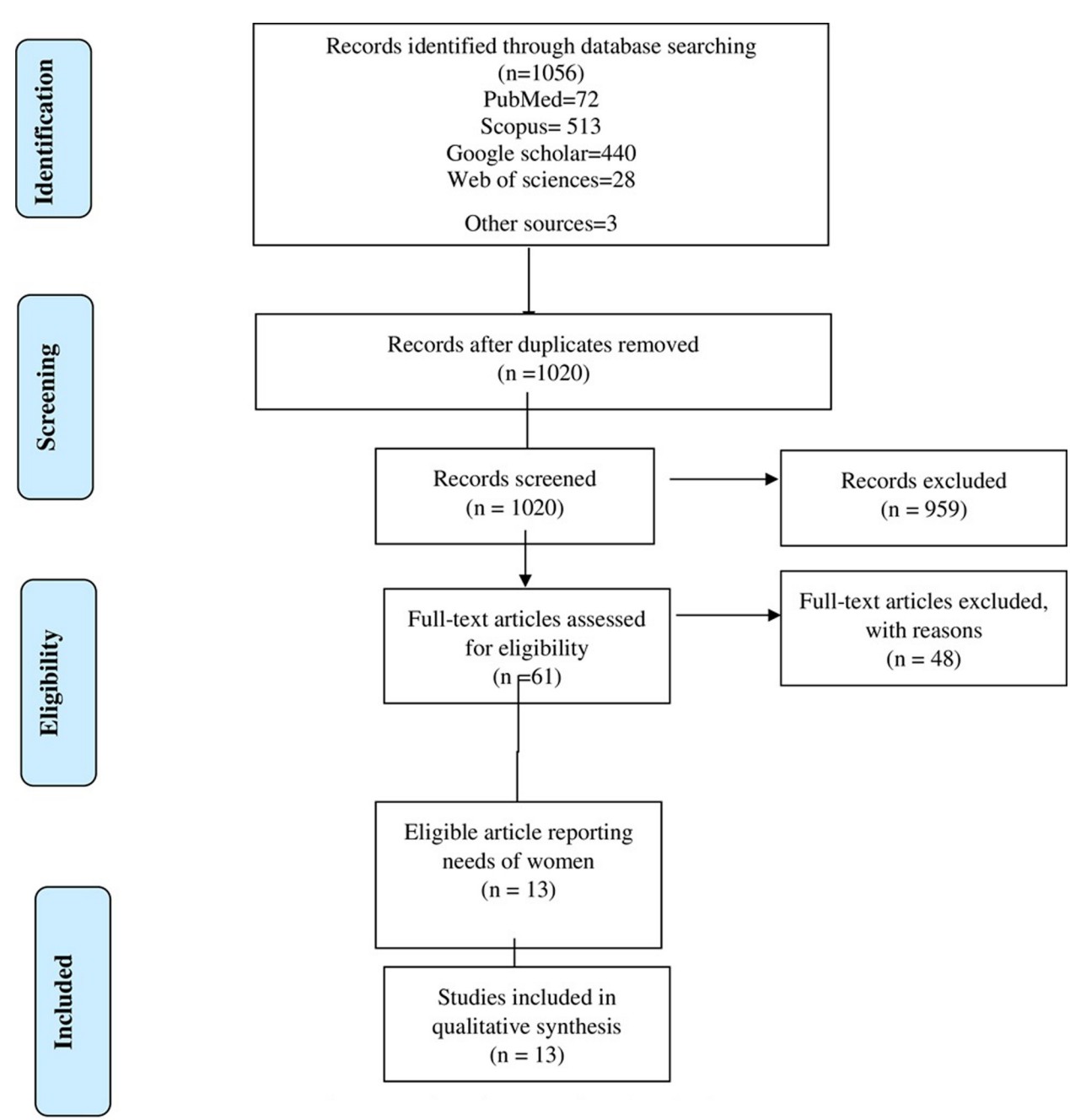

**Fig 1. Flow diagram of study selection.**

for fertility and pregnancy related information, 6. HPV screening and cervical cancer and vaccination.

**1. Signs, symptoms, transmitted, cause, consequences of HPV.** People with the cause of HPV and how it is transmitted, the signs and symptoms of HPV infection and cervical cancer, how to get screened for cervical cancer, the consequences of the virus and its effects (*for example, does the virus always cause cervical cancer?*), *Timeline (for example, how long does HPV cause cervical cancer?), Control and treatment (can it be prevented?), Information on how long the infection stays in the body, And does it lead to cancer?*

**Table 1. Characteristics of studies in the review.**

| No | Reference | Country | Aim | Participants (n) | Study design | Analysis | Disclosure outcomes report |
|---|---|---|---|---|---|---|---|
| 1 | Sophie Mulcahy Symmons, et al. 2021 [28] | England | This study aimed to investigate women's information needs and suggestions for improving communication following receiving a positive HPV test result among people with higher and lower levels of education. | • 30 women<br><br>• age 24 to 63<br><br>• HPV positive | • In-depth semi-structured interview<br><br>• Face-to-face interviews<br><br>• 15 women with highly educated and 15 with low-educated women.<br><br>• The women participated in the interview on average 35.5 days after receiving their HPV test results. | Data were coded using qualitative analysis software NVivo 12 | Summary of information needs in general and among people with higher and lower education including:<br><br>**Get test results using letters**<br><br>✓ Not knowing the test result<br><br>✓ before receiving the test result Poor knowledge about HPV<br><br>✓ Try and search to interpret the test result<br><br>**Content and structure of the received letter**<br><br>✓ Request more information about HPV<br><br>✓ Request personalization of the sent letter<br><br>✓ Prioritize the test result by first mentioning the natural cytological result and then the HPV-related result<br><br>✓ Explain about the next follow-up and additional tests<br><br>**Search for information after receiving the test result**<br><br>✓ Talk to your doctor about the test result<br><br>✓ Search for information from cyberspace<br><br>✓ Receive appropriate information along with providing test results<br><br>✓ Introducing trusted websites<br><br>• In the group of people with higher education, people preferred to receive Supportive information. But in the group of people with lower education in terms of understanding and receiving information<br><br>**Questions about test interpretation**<br><br>✓ Be explicitly informed about the risk of uterine cancer.<br><br>✓ Ask for information about the cause of HPV and how it is transmitted<br><br>✓ The effect of HPV on gender and sexual partner<br><br>✓ The effect of HPV on fertility<br><br>✓ Lack of clear association between normal cytology and HPV positive test<br><br>✓ Lack of a clear link between HPV and other sexually transmitted diseases<br><br>✓ How to share the test result with your partner and sexual partner |
| 2 | Laura Marlow, et al. 2020 [40] | England | The aim of this study was to investigate more information needs among women undergoing primary HPV screening. | • 381 women<br><br>• Aged 24–65 who have registered one or more responses. | • Qualitative study<br><br>• The postal questionnaire was completed immediately, 6 and 12 months after receiving the results of the human papillomavirus (HPV) test.<br><br>• Ask the participant an open-ended question entitled "Do you have an unanswered question about cervical screening and HPV testing?" Was asked. | content analysis | A total of 921 women who underwent initial HPV screening returned their original questionnaire. A total of 507 free texts were registered (base = 239/931, 6 months = 762/110 and 12 months = 537/68). Results as:<br><br>Reaction and understanding of test results (45%) 170<br><br>✓ Emotional response<br><br>✓ Trust the results<br><br>✓ Meaning of results<br><br>✓ Impact on sexuality<br><br>✓ Lack of general understanding<br><br>✓ Epidemiology of HPV<br><br>✓ Cause of HPV<br><br>✓ Prevention / treatment of HPV<br><br>✓ HPV vaccination |

*(Continued)*

**Table 1.** (Continued)

| No | Reference | Country | Aim | Participants (n) | Study design | Analysis | Disclosure outcomes report |
|---|---|---|---|---|---|---|---|
| 3 | Qaderi, et al. 2020 [41] | IRAN | The aim of this study was to evaluate the needs and experiences of HPV positive women receiving health services. | 40 HPV positive women | semi-structured interviews | content analysis | The subject and subcategory of this study include: |
| | | | | | | | **1. Communication and counseling skills** |
| | | | | | | | A. Discussion about HPV |
| | | | | | | | B. Emotional support and acceptance of the disease |
| | | | | | | | C. Provide recommendations |
| | | | | | | | D. Clinical considerations |
| | | | | | | | **2. Commitment to professional principles** |
| | | | | | | | A. Gain the trust of patients |
| | | | | | | | B. Avoid financial misconduct |
| | | | | | | | C. Assertiveness and privacy |
| | | | | | | | D. Clinical considerations |
| | | | | | | | 3. **Providers' knowledge about HPV** |
| | | | | | | | A. Follow the screening instructions |
| | | | | | | | B. Avoid misconceptions |
| | | | | | | | C. Adopt multilateral approaches |
| | | | | | | | Other needs and questions of patients include: |
| | | | | | | | ✓ Search for information on HPV types, transmission, spread and removal of the virus from the body |
| | | | | | | | ✓ Need to request information about other sexually transmitted diseases, the potential benefits of the HPV vaccine, and the risk of HPV-related cancers |
| | | | | | | | ✓ Questions about HPV cofactors for cervical cancer in women with abnormal cytology |
| | | | | | | | ✓ Participants look for ways to improve their safety and sexual health |
| 4 | Kathrin Milbury et al. 2013 [31] | U.S.A | The aim of this study was to evaluate the information and psychosocial needs of patients with oral and pharyngeal squamous cell carcinoma in HPV-positive individuals and to identify the psychosocial challenges associated with HPV-positive cancer. | HPV men and women are positive with the OPSCC and have relationships | ▪ Exploratory study | analyzed using SPSS v19 | The results of this study include: |
| | | | | | ▪ Use open-ended questions such as "What information and knowledge would you like to know about HPV and related cancer?" | | ✓ 66% of patients were aware of their HPV status but only 35% of them considered HPV to be a possible cause of cancer. |
| | | | | | ▪ Questionnaires used include: Gritz 8-item questionnaire to assess past and present nicotine use and drug interactions | | ✓ Most patients report their HPV status to their partner. |
| | | | | | ▪ 10 item questionnaire (AUDIT) consumption assessment | | ✓ 41% discussed virus transmission and only 23% were aware of potential risks and preventive measures. |
| | | | | | | | ✓ 39%of patients sought to talk to an oncologist for information on HPV issues, and 58% sought it from other sources. |
| | | | | | | | ✓ More than a third of them were interested in learning more about HPV. |
| | | | | | | | ✓ Women are interested in receiving any information. 18% want more information on how HPV causes cancer. 15% wanted information on HPV vaccination (especially whether vaccinating their children to prevent cancer), 10% wanted information on how to prevent transmission to their partner, and 10% wanted to know if there was any cure for Is there HPV or not? |
| | | | | | | | The findings suggest that the diagnosis of HPV + OPSCC may have important psychosocial consequences including stigma, self-blame, and social interaction problems |

(*Continued*)

**Table 1.** (Continued)

| No | Reference | Country | Aim | Participants (n) | Study design | Analysis | Disclosure outcomes report |
|---|---|---|---|---|---|---|---|
| 5 | Dorothy Gold, et al. 2012 [48] | U.S.A | The aim of this study was to understand a framework for describing and recognizing the psychosocial factors and challenges that HNC patients face, including those with HPV-related tumors during their disease stages from diagnosis, treatment, and recovery to survival, it shows. | - | Qualitative Study | - | ✓ Women with HPV need psychosocial care at different stages of the disease. |
| | | | | | | | ✓ Shock, uncertainty, and fear of deformed limbs following genital warts, dysfunction, or disability. Patients with HNC face physical and psychological challenges during illness. |
| | | | | | | | ✓ There is a high potential for psychosocial disorders, including anxiety and depression, during ongoing care, including survival. |
| | | | | | | | ✓ Patients with HNC associated with human papillomavirus (HPV) are at risk for psychological stress due to their demographic characteristics as well as the viral cause of their tumors. |
| | | | | | | | ✓ Failure to meet psychosocial needs can complicate the course of treatment and recovery. |
| 6 | SungJong Lee1, et al. 2012 [36] | Korea | Identify the questions that women ask on the human papillomavirus-related website. (www.hpvkorea.org) | Of the 3062 subjects who visited the HPV website, 2330 asked general questions and 732 asked private questions. | Comparative Study | Chi-square test. | ✓ 10 categories for topics and sample questions with different topics are shown in. |
| | | | | | Data on all questions raised between March 2004 and July 2011 were collected and analyzed. | Statistical analysis was performed using SAS program, version 8.0. | ✓ Of the 3,062 people who visited the HPV website, 2,330 asked general questions and 732 asked private questions. |
| | | | | | All questions on the HPV website were in Korean and in free text. | P <0.05 values were defined as significant. | |
| | | | | | At the time of drafting, the text was being translated from Korean to English. | | ✓ The type and frequency of public and private questions showed a statistical difference between the two groups (p <0.001). |
| | | | | | Data on all questions raised between March 2004 and July 2011 were collected and analyzed. | | ✓ From 2004 to 2011, the most common questions are as follows: Frequency and percentage of each question are as follows. |
| | | | | | All questions on the HPV website were in Korean and in free text. | | |
| | | | | | At the time of drafting, the text was being translated from Korean to English. | | **1. Treatment of HPV and cervical dysplasia 1156 (37.8)** |
| | | | | | | | Is HPV treatable? What treatment is recommended for HPV infection? |
| | | | | | | | What is the best treatment for my condition (for example, cervical dysplasia?) |
| | | | | | | | **2. How HPV is transmitted 684 (22.3)** |
| | | | | | | | Ask for information about the routes of transmission and duration of HPV infection |
| | | | | | | | Do other people become infected with HPV by sharing towels or bowls of soup? |
| | | | | | | | Have I got HPV infection from my partner? |
| | | | | | | | Does skin warts spread to other parts of the body than to the genitals? |
| | | | | | | | Is genital warts transmitted to the baby during breastfeeding? |
| | | | | | | | **3. HPV regression 481(15.7)** |
| | | | | | | | How long does it take for an HPV infection to go away on its own? |
| | | | | | | | Does a negative HPV DNA test really indicate that the HPV virus has been removed from the body? |
| | | | | | | | **4. Genital warts 174(5.7)** |
| | | | | | | | Why do genital warts recur even after successful treatment? |
| | | | | | | | If a small lump is found in the genital area, is it a genital wart? |
| | | | | | | | **5. Risk factors for cervical cancer 147(4.8)** |
| | | | | | | | What is the risk of cervical cancer after HPV infection? |
| | | | | | | | What is the risk of cervical cancer due to cervical dysplasia? |

(*Continued*)

**Table 1.** (Continued)

| No | Reference | Country | Aim | Participants (n) | Study design | Analysis | Disclosure outcomes report |
|---|---|---|---|---|---|---|---|
| | | | | | | | **6. HPV vaccination 116(3.8)** |
| | | | | | | | **How effective is HPV** vaccination? |
| | | | | | | | Which company vaccine is most effective in preventing HPV? |
| | | | | | | | When can the vaccine be used? |
| | | | | | | | **7. Prevention of HPV 90(2.9)** |
| | | | | | | | **How to prevent HPV** infection? |
| | | | | | | | Are Diets Good for HPV Prevention? |
| | | | | | | | Can condoms always prevent HPV infection? |
| | | | | | | | **8. Vaginal discharge 86 (2.8)** |
| | | | | | | | **Is vaginal discharge** associated with symptoms of HPV infection? |
| | | | | | | | **9. HPV and pregnancy 81(2.6)** |
| | | | | | | | Is HPV transmitted from mother to baby during pregnancy? |
| | | | | | | | **10. Laryngeal cancer and HPV 47(1.5)** |
| | | | | | | | If a lump or lump is felt in the mouth, is this laryngeal cancer associated with HPV? |
| 7 | Sandra Millon Underwood, et al. 2010 [29] | U.S.A | The aim of this study was to obtain the required information about HPV, cervical cancer and condom use. | 4 single women | cross-sectional, mixed methods, exploratory survey design was used in this study | Content analysis for qualitative study | Women age 18 to 44 years. |
| | | | | | | | Sexual behavior, chronic infections, HPV, are risk factors for cervical cancer. |
| | | | | | | Quantitative data analysis of SPSS using descriptive and inferential statistics | 76.8% (352) of women reported having had sexual activity in the last six months. Among them, 62.2 (219) reported having one sexual partner, 15.6% (55) reported having two sexual partners and 22.2% (78) reported having three or more partners in the past six months. |
| | | | | | | | Floors and basements include: |
| | | | | | | | **Cause of HPV** |
| | | | | | | | Where does HPV come from? How does one actually get it? How many different types of HPV are there? Do all types of HPV cause cancer? Do some types of HPV cause warts? |
| | | | | | | | How is HPV diagnosed in a sexual partner? What are HPV lesions like? |
| | | | | | | | Why cervical cancer is called a sexually transmitted disease? |
| | | | | | | | Can I get HPV infection without having sex? |
| | | | | | | | Can I get HPV from contact with strangers? |
| | | | | | | | Is HPV transmitted from person to person by kissing? Can it be transmitted to the throat? |
| | | | | | | | Who is most at risk for HPV infection? |
| | | | | | | | Does having more than one sexual partner increase a person's risk of contracting HPV? |
| | | | | | | | Does HPV increase with sexual activity? |
| | | | | | | | Where does the infection enter the body? |
| | | | | | | | **Cause of cervical cancer** |
| | | | | | | | What is the main cause of cervical cancer? |
| | | | | | | | Is cervical cancer inherited? If we do not have a family history, what is the cause of this disease? |
| | | | | | | | Who is most at risk for cervical cancer? |
| | | | | | | | Does having more than one sexual partner increase the risk of cervical cancer? |
| | | | | | | | What is cervical cancer like? |
| | | | | | | | Does anyone diagnosed with HPV infection get cervical cancer? |
| | | | | | | | **Signs and symptoms of HPV infection** |
| | | | | | | | How do we know if we have been exposed to HPV or become infected with HPV? |
| | | | | | | | What are the signs and symptoms of HPV infection? |

*(Continued)*

**Table 1.** (Continued)

| No | Reference | Country | Aim | Participants (n) | Study design | Analysis | Disclosure outcomes report |
|----|-----------|---------|-----|------------------|--------------|----------|----------------------------|
| | | | | | | | **Signs and symptoms of cervical cancer** |
| | | | | | | | What are the symptoms of cervical cancer? Are there any early warning signs? |
| | | | | | | | Does cervical cancer cause pain? |
| | | | | | | | **HPV and cervical cancer screening** |
| | | | | | | | How easy is it for doctors to diagnose cervical cancer? |
| | | | | | | | Is a Pap smear really enough to diagnose? |
| | | | | | | | Can HPV be detected before cancer develops? |
| | | | | | | | In addition to annual examinations, what kind of examination should we do? |
| | | | | | | | **HPV vaccine** |
| | | | | | | | Is there a vaccine to prevent HPV? |
| | | | | | | | How effective is the vaccine? |
| | | | | | | | Is the vaccine safe? |
| | | | | | | | Why is the vaccine so expensive? |
| | | | | | | | Where can I go to get the vaccine in the community? |
| | | | | | | | Where can I get more information about the HPV vaccine? |
| | | | | | | | **HPV and cervical cancer risk management** |
| | | | | | | | What can be done to reduce the risk of HPV infection and cervical cancer? |
| | | | | | | | What can we do to prevent girls and younger women from being exposed? |
| | | | | | | | What can older women do to reduce their risk of HPV and cervical cancer? |
| 8 | Veronique Verhoeven, et al. 2010 [30] | Belgium | Determine the specific information needs of people searching the Internet for information on human papillomavirus (HPV). | 527 emails were received in which 713 separate questions were identified and answered. | Data on all questions raised between March 2004 and July 2011 were collected and analyzed. All questions on the HPV website were in Korean and in free text. At the time of writing the manuscript, the text was translated from Korean to English. Website address: (www.ua.ac.be/hpvinfo) | qualitative content analysis | All questions are categorized in one of the following topics. In descending order: • transfer (n = 162) • Vaccine (n = 98) • Natural history (n = 70) • Defective circle (n = 60) (meaning: the belief that two) Partners with HPV infection continue to infect each other, so it is not possible to clear the virus. • Diagnosis of HPV in men (n = 52) • Diagnosis of HPV in women (n = 41) • Treatment in women (n = 40) • Commune period (n = 34) • Pregnancy / fertility (n = 27) • Genital warts (n = 23) • Loyalty and unfaithfulness (n = 21) • Treatment in men (n = 19) • Symptoms of HPV infection (n = 10) 1. When did I get the virus? when? Did my partner take it from a former relationship or betray me? 2. Is the virus transmitted through towels, dirty toilet seats, pets? 3. Can children be infected with HPV? 4. What can be done to prevent HPV transmission? 5. When diagnosed with HPV in a couple. What are the consequences for his wife and sexual partner? Do you have a sexual partner? Should he test? 6. I have HPV and my husband will probably get it too. Won't we constantly infect each other? |

(*Continued*)

**Table 1.** (Continued)

| No | Reference | Country | Aim | Participants (n) | Study design | Analysis | Disclosure outcomes report |
|---|---|---|---|---|---|---|---|
| | | | | | | | 7. When I got HPV. What to say to my partner, ex-partner, and my lover. To? |
| | | | | | | | 8. I have HPV. Does this fact affect my sex life? |
| | | | | | | | 9.Does HPV infection jeopardize my chances of getting pregnant? |
| | | | | | | | 10. I just got my first HPV vaccine. Should I abstain from sex until after the third injection? Can I have sex while using a condom? |
| | | | | | | | 11. I was diagnosed with HPV. Is vaccination good for me? |
| 9 | Laura A V Marlow, 2009 [42] | British | This study aimed to identify key questions about HPV | • 21 women<br>• 18–53 yesrs old | • Face-to-face interview<br>• The samples were entered into the study using the snowball technique | thematic framework analysis | The questions were in 6 themes: |
| | | | | | | | **Identity (for example, what are the symptoms?)** |
| | | | | | | | Does HPV have symptoms? |
| | | | | | | | Is it really common in the UK? |
| | | | | | | | **Cause (for example, how do you get HPV?)** |
| | | | | | | | How to fight HPV? |
| | | | | | | | Can HPV be transmitted in a way other than sexually? |
| | | | | | | | Is HPV more common in young people? |
| | | | | | | | Are women who are multiple partners and sex workers more at risk? |
| | | | | | | | Does drinking alcohol increase the risk of infection? |
| | | | | | | | **Schedule (for example, how long does it take?)** |
| | | | | | | | How long does it take for HPV to become cervical cancer? |
| | | | | | | | How long can the virus remain dormant? |
| | | | | | | | **Consequences (for example, does it always cause the cervix?)** |
| | | | | | | | How many cases of HPV become cervical cancer? |
| | | | | | | | Is HPV the only cause of cervical cancer? |
| | | | | | | | What other cancers are caused by this virus? |
| | | | | | | | Can I get pregnant with this virus? |
| | | | | | | | Does HPV have other effects on the body? |
| | | | | | | | Are Genital Warts and Cancer Related? |
| | | | | | | | How many women get cervical cancer? |
| | | | | | | | How is cervical cancer treated with treatment? |
| | | | | | | | Can a man get HPV? |
| | | | | | | | Does it affect men? |
| | | | | | | | Should the sexual partner be notified if the HPV test is positive? |
| | | | | | | | **Treatment control (e.g. Can you prevent infection?)** |
| | | | | | | | Is there a way to protect yourself against HPV? |
| | | | | | | | Does using a condom protect against the virus? |
| | | | | | | | What is the treatment for HPV? |
| | | | | | | | What does the treatment of cellular changes involve? |
| | | | | | | | How long does it take for this virus to clear the body? |
| | | | | | | | Why do some people clean it with their immune system, but others do not? |
| | | | | | | | How often should you go for a checkup? |
| | | | | | | | At what age should screening for the virus begin? |
| | | | | | | | **Methods: HPV testing/ vaccination** |
| | | | | | | | What does the [HPV] test include? |
| | | | | | | | Can I get an HPV test from a doctor? |
| | | | | | | | How long does it take to determine the result of an HPV test? |
| | | | | | | | Is HPV included in the STI test? |

(*Continued*)

**Table 1.** (*Continued*)

| No | Reference | Country | Aim | Participants (n) | Study design | Analysis | Disclosure outcomes report |
|---|---|---|---|---|---|---|---|
| | | | | | | | Is Pap smear different from HPV? |
| | | | | | | | How effective is the vaccine? |
| | | | | | | | What are the side effects of the vaccine? |
| | | | | | | | What are the long-term effects of the vaccine? |
| | | | | | | | How does the vaccine work, is it a bit of a virus? |
| | | | | | | | Can the virus be vaccinated? |
| | | | | | | | Is the vaccine for prevention or treatment? |
| | | | | | | | How many vaccinations should be given? |
| | | | | | | | How many years have they been working on the vaccine? |
| | | | | | | | Who has the vaccine been tested on? |
| | | | | | | | Who has tested the vaccine? |
| | | | | | | | Will the vaccine be suitable for everyone? |
| | | | | | | | Is the vaccination only for a certain age group? |
| | | | | | | | When is the vaccine available? |
| | | | | | | | Will the vaccination be free for everyone? |
| | | | | | | | Does the vaccine also protect against warts? |
| | | | | | | | ↳ Should I have a Pap smear and test after the injection? |
| 10 | Allison L et al. 2008 [50] | African-American | This study aims to understand the results of HPV and Pap smear test entitled: What do women with HPV need and want to know? Was performed | ▪ A total of ten focus groups (N = 90)<br>▪ Age<br>▪ 30–45 years; 46–65 years<br>▪ Language English/ Spanish | ▪ Qualitative Study<br><br>▪ Participants' knowledge about cervical cancer and HPV and the reactions of each group were recorded with more than 2 observers / notes. | The results were analyzed using a note-based strategy with three independent judges. | According to the study, participants were aware of HPV and cervical cancer, but did not have the necessary knowledge. |
| | | | | | | | They were confused about the nature of the disease, the severity, the transmission, the types, the treatment, and the instructions for the HPV test, and needed to have complete information because they stated that this reduced their anxiety and caused them to follow up. Do the necessary. |
| | | | | | | | Many questions about your sexual partner and how to deal with it If you have the virus, several concepts of HPV for women are contradictory and must be carefully explained to ensure that patients understand their test results. |
| | | | | | | | Do not experience unnecessary fear, anxiety or stigma; And return for follow-up. Recommendations are provided to meet these patient needs. |
| 11 | Suzanne M. et al. 2006 [37] | Australia | Purpose, how to manage and communicate with patients about HPV? | - | - | - | The needs of the community in relation to HPV |
| | | | | | | | ✓ What is HPV and how do we get it? |
| | | | | | | | ✓ How long does the infection stay in the body and does it lead to cancer? |
| | | | | | | | ✓ How is HPV diagnosed? |
| | | | | | | | ✓ Despite a normal Pap smear, is there no cure for the virus? |
| | | | | | | | ✓ Who should do the HPV DNA test and when? |
| | | | | | | | ✓ HPV can cause cancer: Are other cancers caused by these microorganisms? |
| | | | | | | | ✓ Does taking OCP pills affect this disease? Should it be stopped? How long does taking the pill cause uterine cancer? |
| | | | | | | | ✓ I have had this partner for many years. Now that I'm positive for HPV, does that mean she've had sex with someone else? |
| | | | | | | | ✓ Is there a difference between a low-risk and a high-risk type? |
| | | | | | | | ✓ Do I have to tell my partner? |
| | | | | | | | ✓ Do I have to quit smoking? |
| | | | | | | | ✓ If my spouse and I were in the beginning of a virgin relationship, how can I have an abnormal HPV or Pap smear? |

(*Continued*)

**Table 1.** (Continued)

| No | Reference | Country | Aim | Participants (n) | Study design | Analysis | Disclosure outcomes report |
|----|-----------|---------|-----|------------------|--------------|----------|----------------------------|
| 12 | KirstenMcCaffery, et al. 2005 [34] | Australia | The aim of this study was to determine the information needs of women about the role of human papillomavirus (HPV) in cervical cancer and how to prevent cervical cancer. | 19 women with HPV infection were selected after routine cervical screening. | Qualitative face-to-face interview | Framework Analysis | 1. Women's perception of HPV<br>✓ Knowledge of HPV before current diagnosis<br>✓ The patient's current understanding of HPV<br>2. Required information about HPV<br>✓ High-risk types of HPV against genital warts<br>✓ Sexual transmission: Where does the virus come from?<br>✓ Impact on sexual partner, future partners and disease transmission between partners<br>✓ Outbreak of HPV, information on recurrence and recovery<br>✓ Information on management and treatment options<br>✓ Consequences of cancer risk and fertility<br>✓ Lack of appropriate and available information<br>✓ Key information that women reported as reliable information<br>✓ High prevalence of HPV<br>✓ How the disease goes away on its own<br>✓ Creating cancer following infection with the virus<br>✓ Absence of genital warts and symptoms of the disease<br>3. How to present HPV results<br>✓ Using the phone<br>✓ Using letters<br>✓ During the consultation<br>4. Physician communication method<br>✓ Notification style<br>✓ Amount of information<br>5. HPV information retrieval experiences |
| 13 | Rebecca Anhang, et al. 2004 [17] | USA | Determining the relevance of HPV: Reviewing existing research and recommendations for educating patients | - | Qualitative Study | - | According to this study, information and psychological needs were among the important needs of individuals, which include:<br>✓ How and when did I get HPV?<br>✓ Does HPV affect pregnancy or the baby?<br>✓ Can a person get HPV orally or by hand?<br>✓ How can I get tested for HPV?<br>✓ Do I always have HPV?<br>✓ How can HPV be prevented from being given or received?<br>✓ Can partners re-infect each other?<br>✓ Does HPV cause cervical cancer?<br>✓ What should I tell my partner about HPV?<br>✓ What are the best HPV treatment options?<br>Looking for psychological problems such as:<br>▪ Anxiety<br>▪ Regret<br>▪ Angry<br>▪ Fear of cancer<br>▪ Concerns about loss of reproductive functions<br>▪ Fear of negative reactions from friends, family or sexual partners<br>▪ Concerns about partner infidelity or enmity with a person are thought to be a source of infection<br>▪ Changes in body image<br>▪ Reduce intimate activities |

**Table 2. CASP (Critical Appraisal Skills Programme) qualitative checklist scores for total included studies.**

| | CASP criterion | | | | | | | | | | Total Score | Quality of study |
|---|---|---|---|---|---|---|---|---|---|---|---|---|
| | Clear statement of aim | Suitable qualitative methodology | Appropriate research design | Proper recruiting strategy | Adequacy in data collection | Adequate relationship between researcher and participants | Ethical considerations | Rigor in data analysis | Clear statement of findings | Overall value of research | | |
| Sophie Mulcahy Symmons, et al. 2021 | green | green | yellow | green | green | green | green | green | green | green | 19 | Moderate |
| Laura Marlow, et al. 2020 | green | green | green | green | green | green | green | green | green | green | 20 | High |
| Qaderi, et al. 2020 [41] | green | green | green | green | green | green | green | green | green | green | 20 | High |
| Dorothy Gold, et al. 2012 | yellow | yellow | yellow | green | green | red | green | green | green | yellow | 14 | Low |
| Veronique Verhoeven, et al. 2010 | green | green | green | green | green | green | green | green | green | green | 20 | High |
| Laura A V Marlow, 2009 | green | green | green | green | green | green | green | green | green | green | 20 | High |
| KirstenMcCaffery, et al. 2005 | green | green | green | green | green | green | green | green | green | green | 20 | High |
| Rebecca Anhang, et al. 2004 | green | yellow | yellow | green | green | green | yellow | green | green | green | 17 | Moderate |

**2. HPV and sexual partner.** Women with HPV need counseling on how to communicate and deal with a sexual partner. Because some women expressed concern about their husbands' infidelity and a lack of trust was expressed between the couple [28].

*When I get HPV, how do I tell my partner what to say*? *What effect does HPV have on sexual life*? Because sometimes these issues lead to uncertainty and stress and in severe cases lead to the breakdown of relationships [29].

The most common question asked by women with HPV was: *which sexual partner (active or former) did the infection come from*? *Did my partner get it from a previous relationship or did he or she betray me*? *When HPV is diagnosed in a couple, what are the consequences for the spouse and their sexual partner*? *Will her sexual partner get it too*? *Should her partner test and follow up*? *When a couple is infected with HPV and is being treated, will they get the virus again from their partner*? *Will they constantly infect each other*? *Will they use a condom*? *Can I have sex*? *Can a person get HPV orally or by hand*? [30, 31].

**3. Seeking needed information for women with HPV.** Because many women with the condition felt that they did not have enough information about HPV, many sought more information, including the Internet (social media, blogs, and websites provided by labs and private professionals) [32]. Women with HPV patients think that finding reliable and up-to-

**Table 3. CASP (Critical Appraisal Skills Programme) crossectional checklist scores for total included studies.**

| | CASP criterion | | | | Total Score | Quality of study |
|---|---|---|---|---|---|---|
| | Profile of participants | Evaluation tool | Study design | Results | | |
| Kathrin Milbury et al. 2013 | green | green | green | green | 18 | High |
| SungJong Lee1, et al. 2012 | green | green | green | green | 18 | High |
| Sandra Millon Underwood, et al. 2010 | green | green | green | green | 18 | High |

date information is challenging. Most HPV patients prefer the information provided by official websites. Women who were regular internet users reported finding useful information [33].

However, it was often not easy for them to interpret and determine the correctness or incorrectness of the material. Women who were regular Internet users reported finding useful information [28]. However, it was often not easy for them to interpret and determine the correctness or incorrectness of the material. Women reported that most cases of finding information about HPV, along with the effect of the virus on genital warts and other sexually transmitted diseases, in which risk factors such as having multiple sexual partners were highlighted, some women were very upset and experienced a sense of shame and embarrassment [34]. Or, in many cases, this information was in addition to information related to uterine cancer, which led to fear, anxiety, and stress in them [28]. It seems that the information that women are looking for and the way their doctor informs them can affect their psychological response to HPV infection and is one of the issues that is considered important in follow-up [35].

**4. HPV and social consequences.**   Dysfunction or inability to perform daily activities, limitation in relationships with others, limitation of social activities, shock, insecurity, and fear of deformity of the limbs following the development of genital warts and its impact on quality of life were among the challenges [36]. Women with HPV experienced it during the illness. Therefore, women with HPV need psychosocial care at different stages of the disease, so that non-fulfillment of psychosocial needs can complicate the course of treatment and recovery [37].

**5. Needs for fertility and pregnancy related information.**   Women with HPV raised some issues related to fertility, and the need for proper counseling on the impact of the virus on reproductive health was felt [32].

Many women wanted information about the meaning of HPV infection in relation to the increased risk of cervical cancer [38]. They also wanted to know what they could do about it themselves and whether HPV could impact on their fertility, the effect of HPV treatments and vaccinations' effects on fertility, fetal abnormalities and adverse pregnancy outcomes (e.g., abortion and Preterm delivery) [39].

Some women postpone the decision to become pregnant until the cytology results improve because for fear of not being properly diagnosed with the disease during pregnancy or the side effects of treatment and vaccination. Therefore, these women need to know what to do to protect their reproductive health and reduce its complications against infection [40].

Since genital warts can multiply during pregnancy, removing warts during pregnancy was one of the most important issues. Women with genital warts have revealed the need for information about the teratogenicity of some wart removal treatments. They were anxious about any threat it might pose to the fetus [36].

Information on contraception, fear of premature menopause, fear of cervical cancer, fear of hereditary cancer caused by HPV were other issues raised in this area [40].

To prevent unplanned pregnancies, women with HPV need more information to choose the preferred method of contraception. Because long-term use of birth control pills increases the risk of cervical cancer for women with persistent HPV, they need guidance in changing their method of contraception. Women sought information on the negative effects of combined oral contraceptive pills (COCs) and levonorgestrel (LNG) pills on their cellular changes [41].

**6. HPV screening and cervical cancer and vaccination.**   Another piece of information that infected women needed to know about was screening. They needed to know how helpful these screenings can be in diagnosing cervical cancer [36]. These include: "*How is screening done? Is a Pap smear enough to diagnose? Can HPV be diagnosed before cancer develops? In*

*addition to annual checkups, what type of checkups are started at what intervals and at what age? What does the [HPV] test include? Can I get an HPV test from a doctor? How long does it take to determine the result of an HPV test? What is the difference between an HPV test and a Pap smear?"* [29]

The women also asked for information about the effect of the vaccine, side effects, long-term effects, mechanism of action of the vaccine, number of doses received, and the age range for vaccination [42].

## Quantitative studies

Lee, *et al.* (2012), In their quantitative study, it showed 10 categories for topics and sample questions about different topics. Of the 3,062 people who visited the HPV website, 2,330 asked general questions and 732 asked private questions. The type and frequency of public and private questions showed a statistical difference between the two groups (p<0.001).

From 2004 to 2011, the most common questions, respectively: a. Treatment of HPV and cervical dysplasia (37.8%) 1156, b. How HPV transmission (22.3%) 684, c. HPV regression (15.7%) 481, d. Genital warts (5.7%) 174, e. Risk factors for cervical cancer (4.8%) 147, f. HPV vaccination (8.3%)116, g. Prevention of HPV 90 (9.2), h. Vaginal discharge (%2.8) 86, i. HPV and pregnancy (%2.6) 81, j. Laryngeal cancer and HPV 47 (%1.5) was [36].

Underwood *et al. (2010)* showed that sexual behavior and chronic HPV infections are risk factors for cervical cancer. Sexual behavior and chronic HPV infections are risk factors for cervical cancer. When asked to describe their sexual activity, 76.8% (352) of women reported having had sexual activity in the past six months. Among them, 2.62% (219) reported having a sexual partner, 15.6% (55) two sexual partners and 22.2% (78) had three or more partners in the past six months. The most common questions included: cause of HPV, cause of cervical cancer, signs, and symptoms of HPV infection, signs and symptoms of cervical cancer, HPV and cervical cancer screening, HPV vaccine, HPV and cervical cancer risk management [29].

Milbury, *et al.* (2013) in their study showed that 66% of patients were aware of their HPV status, but only 35 percent identified HPV as a possible cause of their cancer. Most patients disclosed their HPV status to their partner, 41% talked about transmitting the virus, and only 23% felt aware of the potential risks of transmission and precautions. 39% of their oncologists wanted to discuss more HPV issues, and 58% sought it from other sources. More than a third said they were interested in more information about HPV.

Women are interested in receiving any information. 18% wanted more information on how HPV causes cancer. 15% wanted information on HPV vaccination (especially whether to vaccinate their children to prevent cancer), 10% wanted information on how to prevent transmission to their partner, and 10% wanted to know if there was any cure for Is there HPV or not? [31].

The diagnosis of HPV led to high levels of anxiety among women. Confusion and uncertainty about the key aspects of HPV, the style in which the clinician presented the information and the mode of delivering the result appeared to contribute to distress and concern experienced by women. Seeking information from alternate sources also added to some women's anxiety by presenting information about HPV in the context of other highly stigmatized STIs and highlighting multiple sexual partners as a risk factor for infection.

All these factors seemed to contribute to women's conceptualization of HPV, and consequently their psychological response to the infection. Therefore, women need clear, consistent information about HPV in key areas: HPV types (high risk versus low risk), mode of transmission, Implications for sexual relationships and partners, prevalence, latency and regression of HPV, management and treatment options.

Implications for cancer risk and female reproductive health, Fertility and pregnancy related information [40]

Recently, many researchers have used written surveys, interviews, and other data collection methods to assess knowledge about HPV among women. They found that women have relatively little knowledge of HPV and suggested using more training programs [19]. In addition, many patients become confused after hearing their doctor's description of HPV. As a result, many patients search the Internet for seeking information about HPV, but there are still many questions for them [43, 44].

According to the findings, most people, especially when it comes to frequently asked questions, really want to find out about the whole disease process, from HPV infection to recovery [34]. Since most of the studies studied were qualitative, this study was able to identify a wide range of needs of affected women. The needs of women with HPV lead to fear, anxiety and worry [39]. Their anxiety is largely due to their poor knowledge of the virus. Previous studies have suggested that women are often shocked to learn about the sexually transmitted nature of HPV [45, 46]. Therefore, women who are screened for or exposed to the virus also need quality information about HPV and its role in cervical cancer when faced with an abnormal result.

Physicians potentially play an important role in modulating the effects of diagnosis through how HPV is diagnosed [47]. Therefore, recognizing these needs can help health professionals understand what questions they are expected to answer.

Women most frequently asked questions regarding HPV itself in this sample related to symptoms and prevalence, transmission of the virus, duration of infection, time course of cancer risk and viral infection, health and social consequences (e.g. effects on relationships), fertility risk, fetal development and the effects of assisted reproduction, prevention and treatment for HPV. The questions that our sample of women asked were very similar to McCaffery and Irwig (2005) study [34].

Women's experience of searching the Internet for more information on HPV has been reported as problematic, anxious, and stigmatizing infection in the community, as the information provided is often related to other sexually transmitted infections and multiple sexual partners as a risk factor for Infections are prominent [48]. When people go to a website about HPV, they are looking for "HPV and cervical cancer", "HPV transmission", "HPV cure", "Genital warts" and "risk of cervical cancer." Questions also arose concerning how to prevent catching the infection, how to avoid spreading the infection to partners and the importance of disclosing an HPV infection to future partners. The implications for sex in general were an important area for concern: the risks associated with oral sex were mentioned in particular. Sung-Jong Lee showed in their study that "genital warts" and "HPV and pregnancy" may be considered embarrassing topics [36]. Therefore, these findings can be used to provide informed recommendations for clinical or Internet-based communication with patients and the general public.

Questions about the proper use and effectiveness of male and female preventive methods (eg, condoms) to reduce the risk of HPV and cervical cancer; questions about the safety, availability and cost of the HPV vaccine; and questions about the need for further community education and information. It is similar to Underwood studies [29].

Most women raised the need for clear information about the impact of HPV on fertility and sexual partners. These findings are similar to the Marlow et al., 2020 study [40].

Also, women needed more information about HPV-vaccination effects on fertility. Therefore, women should be informed by healthcare providers that although pregnancy testing is not necessary before the vaccination, the vaccine manufacturers and WHO recommend avoiding HPV vaccination during pregnancy. In cases of unintentional immunization of pregnant women, no intervention is needed [49].

Based on the review, this systematic review is one of the few review studies that comprehensively addresses the needs of women with HPV. The potential advantage of this study is that the HPV-related questions and needs raised by patients were collected. The strength of this study is that it was a comprehensive and systematic review using PRISMA guidelines. In addition, an extensive search strategy was performed without any history or language restrictions. It is possible that due to the range of terms that can be used to describe the needs, some eligible studies may not have been identified in our search. However, we did forward and backward citation research to reduce this possibility. The data was also extracted by one author and reviewed independently by the second author. While qualitative synthesis allows us to identify a range of different factors that help women with HPV. However, since the majority of studies were qualitative, meta-analysis was not possible. It is suggested that future studies should measure the effect of raising awareness about HPV over time to determine whether raising awareness leads to improved health of infected women. Knowing when and how this response to needs is most effective can help determine the most appropriate interventions.

## Conclusion

Surveys showed that the majority of women had unanswered questions about their HPV test results. The information that women thought helped interpret their test results included having a high-risk type of HPV, the risk of short-term and long-term cancer, and cancer survival statistics for the virus. Women also needed information about sexual transmission, how HPV tested positive in a long-term relationship, and the potential consequences for their partners and the risk of re-infection. Younger women had questions about whether HPV could affect fertility. HPV information should be scientifically accurate and minimize any possible stigma that may be associated with the infection due to its sexually transmitted nature. It was found that seeking information from alternative sources also increased women's anxiety by providing information about HPV in the context of other highly sexually transmitted diseases and highlighting several sexual partners as a risk factor for infection.

## Supporting information

**S1 Checklist.**
(DOC)

**S1 File.**
(ZIP)

## Author Contributions

**Conceptualization:** Zahra Motaghi.

**Data curation:** Mina Galeshi, Hoda Shirafkan, Shahla Yazdani, Zahra Motaghi.

**Formal analysis:** Mina Galeshi, Hoda Shirafkan, Zahra Motaghi.

**Methodology:** Mina Galeshi, Hoda Shirafkan, Shahla Yazdani, Zahra Motaghi.

**Project administration:** Mina Galeshi, Zahra Motaghi.

**Resources:** Mina Galeshi, Zahra Motaghi.

**Software:** Mina Galeshi, Hoda Shirafkan, Zahra Motaghi.

**Supervision:** Mina Galeshi, Shahla Yazdani, Zahra Motaghi.

**Validation:** Mina Galeshi, Shahla Yazdani, Zahra Motaghi.

**Writing – original draft:** Mina Galeshi, Hoda Shirafkan, Shahla Yazdani, Zahra Motaghi.

**Writing – review & editing:** Hoda Shirafkan, Shahla Yazdani, Zahra Motaghi.

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
