## [Decision Letter · Decision Letter 0]

10 May 2022

PONE-D-22-08835Reproductive health needs of Human papillomavirus (HPV) Positive women: A systematic reviewPLOS ONE

Dear Author,

Thank you for submitting your manuscript to PLOS ONE. After careful consideration, we feel that it has merit but does not fully meet PLOS ONE’s publication criteria as it currently stands. Therefore, we invite you to submit a revised version of the manuscript that addresses the points raised during the review process.

We look forward to receiving your revised manuscript.

Kind regards,

Halimatus Sakdiah Minhat, DrPH

Academic Editor

PLOS ONE

Journal Requirements:

1. Please ensure that your manuscript meets PLOS ONE's style requirements, including those for file naming. The PLOS ONE style templates can be found at https://journals.plos.org/plosone/s/file?id=wjVg/PLOSOne_formatting_sample_main_body.pdf and https://journals.plos.org/plosone/s/file?id=ba62/PLOSOne_formatting_sample_title_authors_affiliations.pdf. 2. Thank you for stating the following financial disclosure:  [The funders had no role in study design, data collection and analysis, decision to publish, or preparation of the manuscript.] At this time, please address the following queries: a) Please clarify the sources of funding (financial or material support) for your study. List the grants or organizations that supported your study, including funding received from your institution. b) State what role the funders took in the study. If the funders had no role in your study, please state: “The funders had no role in study design, data collection and analysis, decision to publish, or preparation of the manuscript.”c) If any authors received a salary from any of your funders, please state which authors and which funders.d) If you did not receive any funding for this study, please state: “The authors received no specific funding for this work.” Please include your amended statements within your cover letter; we will change the online submission form on your behalf. 3. Thank you for stating the following in your Competing Interests section:  [no competing interests exist].  Please complete your Competing Interests on the online submission form to state any Competing Interests. If you have no competing interests, please state "The authors have declared that no competing interests exist.", as detailed online in our guide for authors at http://journals.plos.org/plosone/s/submit-now  This information should be included in your cover letter; we will change the online submission form on your behalf. 4. In your Data Availability statement, you have not specified where the minimal data set underlying the results described in your manuscript can be found. PLOS defines a study's minimal data set as the underlying data used to reach the conclusions drawn in the manuscript and any additional data required to replicate the reported study findings in their entirety. All PLOS journals require that the minimal data set be made fully available. For more information about our data policy, please see http://journals.plos.org/plosone/s/data-availability. Upon re-submitting your revised manuscript, please upload your study’s minimal underlying data set as either Supporting Information files or to a stable, public repository and include the relevant URLs, DOIs, or accession numbers within your revised cover letter. For a list of acceptable repositories, please see http://journals.plos.org/plosone/s/data-availability#loc-recommended-repositories. Any potentially identifying patient information must be fully anonymized. Important: If there are ethical or legal restrictions to sharing your data publicly, please explain these restrictions in detail. Please see our guidelines for more information on what we consider unacceptable restrictions to publicly sharing data: http://journals.plos.org/plosone/s/data-availability#loc-unacceptable-data-access-restrictions. Note that it is not acceptable for the authors to be the sole named individuals responsible for ensuring data access. We will update your Data Availability statement to reflect the information you provide in your cover letter. 5. PLOS requires an ORCID iD for the corresponding author in Editorial Manager on papers submitted after December 6th, 2016. Please ensure that you have an ORCID iD and that it is validated in Editorial Manager. To do this, go to ‘Update my Information’ (in the upper left-hand corner of the main menu), and click on the Fetch/Validate link next to the ORCID field. This will take you to the ORCID site and allow you to create a new iD or authenticate a pre-existing iD in Editorial Manager. Please see the following video for instructions on linking an ORCID iD to your Editorial Manager account: https://www.youtube.com/watch?v=_xcclfuvtxQ.
 6. Your ethics statement should only appear in the Methods section of your manuscript. If your ethics statement is written in any section besides the Methods, please delete it from any other section.  7. Please ensure that you refer to Figure 1 in your text as, if accepted, production will need this reference to link the reader to the figure. 8. Please review your reference list to ensure that it is complete and correct. If you have cited papers that have been retracted, please include the rationale for doing so in the manuscript text, or remove these references and replace them with relevant current references. Any changes to the reference list should be mentioned in the rebuttal letter that accompanies your revised manuscript. If you need to cite a retracted article, indicate the article’s retracted status in the References list and also include a citation and full reference for the retraction notice.

Reviewers' comments:

Reviewer's Responses to Questions

**Comments to the Author**

1. Is the manuscript technically sound, and do the data support the conclusions?

Reviewer #1: Yes

2. Has the statistical analysis been performed appropriately and rigorously? 

Reviewer #1: N/A

3. Have the authors made all data underlying the findings in their manuscript fully available?

Reviewer #1: Yes

4. Is the manuscript presented in an intelligible fashion and written in standard English?

Reviewer #1: No

5. Review Comments to the Author

Reviewer #1: The article requires language editing. Several parts of article are written incoherently.

Abstract

1. The main findings of the study should be mentioned in the findings. These are not the main findings of this study.

2. The cases mentioned in the conclusion should be included in the findings. In conclusion, the main message of the present study should be mentioned and finally a suggestion.

Method

In the materials and methods section, this study was conducted without language restrictions. Have you studied other languages such as Korean, French, German, etc.? While this criterion for entering this study was English and Persian.

Results:

1. Write a fig.1 in the findings instead of a flowchart.

2. In the form of a trend, the number of articles from top to bottom does not match, for example, how did the number 36 reach 1024 again? The process of selecting articles should be mentioned in detail and with details, for example, how many articles from Pabmed? How many articles from Scopus? Etc

3. In Tables 2 and 3, 11 articles are qualitatively evaluated, while 13 studies are included and selected

discussion

1. The first paragraph of the discussion should state exactly the main results of this study.

2. The findings are repeated in the discussion, while the main findings of this study should be discussed and interpreted. The main findings of this study are about the needs of women infected with papilloma virus and these needs that have been extracted in this study should be discussed and interpreted and addressed.

3. The strengths and limitations of this study should be mentioned at the end of the discussion

---

## [Author Response · Author response to Decision Letter 0]

31 May 2022

Dear Editor-in-Chief 

PLOS ONE

Thank you for your letter and oppertunity to revise our paper. The suggestions offered by the reviewers have been greatly helpful. We have included reviewer’s comments and responded to them individually. We hope the revised manuscript has been improved enough to be suit the PLOS ONE . We are ready for further revision if indicated 

Best wishes

 DR. Zahra Motaghi

Associate professor reproductive health

 Reviewer# 1

1. Is the manuscript presented in an intelligible fashion and written in standard English?

All grammatical errors typos, variations in fonts were cirrdcted throughout the paper.

2. Abstract

We edited

3. Method

In the materials and methods section, this study was conducted without language restrictions. Have you studied other languages such as Korean, French, German, etc.? While this criterion for entering this study was English and Persian.

We edited

Results:

1. Write a fig.1 in the findings instead of a flowchart.

We edited

2. In the form of a trend, the number of articles from top to bottom does not match, for example, how did the number 36 reach 1024 again? The process of selecting articles should be mentioned in detail and with details, for example, how many articles from Pabmed? How many articles from Scopus? Etc

We edited

3. In Tables 2 and 3, 11 articles are qualitatively evaluated, while 13 studies are included and selected

We explain this item

discussion

1. The first paragraph of the discussion should state exactly the main results of this study.

We edited

2. The findings are repeated in the discussion, while the main findings of this study should be discussed and interpreted. The main findings of this study are about the needs of women infected with papilloma virus and these needs that have been extracted in this study should be discussed and interpreted and addressed.

We edited

3. The strengths and limitations of this study should be mentioned at the end of the discussion

We edited

---

## [Decision Letter · Decision Letter 1]

28 Aug 2022

Reproductive health needs of Human papillomavirus (HPV) Positive women: A systematic review

PONE-D-22-08835R1

Dear Dr. motaghi,

We’re pleased to inform you that your manuscript has been judged scientifically suitable for publication and will be formally accepted for publication once it meets all outstanding technical requirements.

Kind regards,

Hugh Cowley

Staff Editor

PLOS ONE

Additional Editor Comments (optional):

Reviewers' comments:

Reviewer's Responses to Questions

**Comments to the Author**

1. If the authors have adequately addressed your comments raised in a previous round of review and you feel that this manuscript is now acceptable for publication, you may indicate that here to bypass the “Comments to the Author” section, enter your conflict of interest statement in the “Confidential to Editor” section, and submit your "Accept" recommendation.

Reviewer #1: All comments have been addressed

2. Is the manuscript technically sound, and do the data support the conclusions?

Reviewer #1: Yes

3. Has the statistical analysis been performed appropriately and rigorously? 

Reviewer #1: Yes

4. Have the authors made all data underlying the findings in their manuscript fully available?

Reviewer #1: Yes

5. Is the manuscript presented in an intelligible fashion and written in standard English?

Reviewer #1: Yes

6. Review Comments to the Author

Reviewer #1: The above item is not applicable and has no item. there is no concerns about dual publication, research ethics, and publication ethics.

7. PLOS authors have the option to publish the peer review history of their article (what does this mean?). If published, this will include your full peer review and any attached files.

Reviewer #1: No

---

## [Editor Report · Acceptance letter]

31 Aug 2022

PONE-D-22-08835R1 

Reproductive health needs of Human papillomavirus (HPV) Positive women: A systematic review 

Dear Dr. Motaghi:

I'm pleased to inform you that your manuscript has been deemed suitable for publication in PLOS ONE. Congratulations! Your manuscript is now with our production department. 

Kind regards, 

on behalf of

Mr Hugh Cowley 

Staff Editor

PLOS ONE